# Decoupling Reasoning and Perception: An LLM-LMM Framework for Faithful Visual Reasoning

## Abstract

Significant advancements in the reasoning capabilities of Large Language Models (LLMs) are now driven by test-time scaling laws, particularly those leveraging extended Chain-of-Thought (CoT) reasoning. Inspired by these breakthroughs, researchers have extended these paradigms to Large Multimodal Models (LMMs). However, a critical limitation emerges: as their reasoning chains extend, LMMs increasingly rely on textual logic, progressively losing grounding in the underlying visual information. This leads to reasoning paths that diverge from the image content, culminating in erroneous conclusions. To address this, we introduce a strikingly simple yet effective training-free visual-reasoning pipeline. The core concept is to decouple the reasoning and perception processes. A powerful LLM orchestrates the high-level reasoning, strategically interrogating a LMM to extract specific visual information required for its logical chain. The LMM, in turn, functions exclusively as a visual question-answering engine, supplying the necessary perceptual details on demand. This lightweight, plug-and-play approach requires no additional training or architectural changes. Comprehensive evaluations validate that our framework effectively governs the visual reasoning process, leading to a significant reduction in visually-unfounded reasoning steps and a substantial improvement in reasoning fidelity.

## 1 Introduction

Recent advances in reasoning-focused Large Language Models (LLMs) such as OpenAI's O1/O3 (o3, 2024) and DeepSeek-R1 (DeepSeek-AI et al., 2025) have demonstrated remarkable capabilities in complex logical reasoning tasks. This progress has rapidly extended to the multimodal domain, with numerous vision-language models emerging to tackle visual reasoning challenges. Models like QvQ (qvq, 2024), Kimi 1.5-V (Team et al., 2025), alongside specialized architectures such as LLaVA-COT (Xu et al., 2025), R1-OneVision (Yang et al., 2025b), VLM-R1 (Shen et al., 2025), and LMM-R1 (Peng et al., 2025), have shown promising results by generating extended chain-of-thought reasoning to solve complex visual problems (Wang et al., 2025). These models integrate visual perception with linguistic reasoning capabilities, enabling them to interpret visual content and perform sophisticated multi-step reasoning across modalities (Li et al., 2025b).

Despite these impressive strides, a critical factor severely impacts the efficacy of current MLLMs in long-chain reasoning: **a progressive loss of visual grounding** (Jung et al., 2025; Liu et al., 2025). *As a reasoning chain extends, the model's inference process tends to rely more on its internal textual logic than on the visual input context, causing its conclusions to diverge from the image content.* This divergence often begins with minor visual inaccuracies that subsequently propagate and compound through the reasoning steps. This can create a cascade of errors, leading to a final conclusion that is inconsistent with the visual facts and undermining the model's reliability in complex tasks. As illustrated in Figure 1, an MLLM might correctly identify a primary object but, further along its reasoning chain, erroneously describe a non-existent attribute, which then fatally flaws subsequent deductions based on this false detail.

To address this limitation, we propose a training-free pipeline based on the principle of decoupling reasoning from perception. This framework delegates these two cognitive functions to distinct, syner-

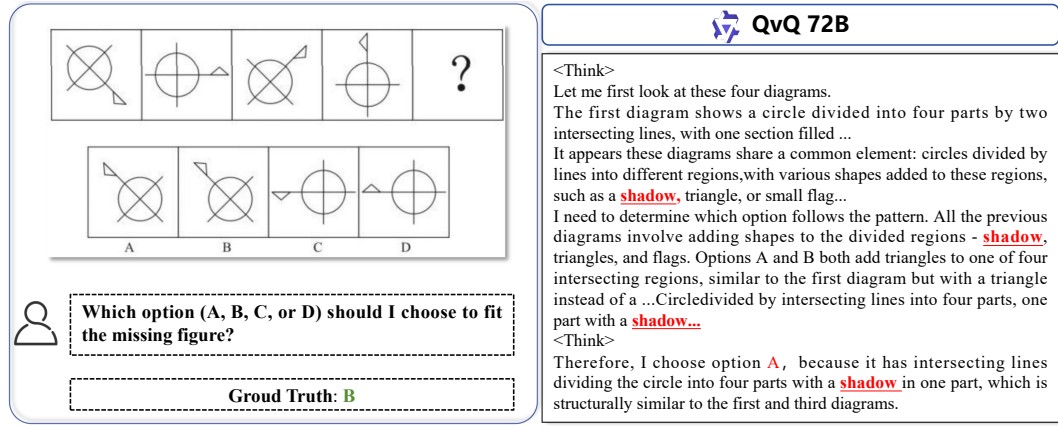

Figure 1: An Illustration of Progressive Loss of Visual Grounding in MLLM-based Reasoning. This figure displays a visual reasoning task where the QvQ 72B model attempts to find the missing pattern. Initially, the model accurately describes the core visual elements. However, as the reasoning chain progresses, it loses its connection to the original visual input and hallucinates a "shadow" to describe the filled quadrant. The model's subsequent deductions are flawed because they are based on this false textual detail.

gistic modules: an LLM-based Reasoner and an LMM-based Observer. The Reasoner, implemented with a text-based LLM, is responsible for the overarching logical inference. The Observer, a standard LMM, is tasked with the sole duty of perception: analyzing the visual input to provide direct answers to the Reasoner's questions.

The operational dynamic is an iterative, structured dialogue. The Reasoner drives the process, methodically interrogating the Observer to extract specific visual details essential to its logical progression. Upon receiving a query, the Observer grounds its analysis in the image and returns a concise answer. This information is then integrated by the Reasoner, which refines its internal state and determines the next line of inquiry. This interactive cycle persists until a conclusion supported by sufficient visual evidence is reached. Significantly, our approach is designed to be model-agnostic, offering a plug-and-play solution that imposes no requirements for architectural changes or costly model fine-tuning.

We evaluate our framework across multiple challenging visual reasoning benchmarks, including MathVision (Wang et al., 2024), MM-Vet (Yu et al., 2024). Experiments show that a Qwen2.5-VL-7B-Instruct model (Bai et al., 2025), directed by a Qwen3-32B model (Yang et al., 2025a), efficiently achieves performance parity with a much larger proprietary model like ChatGPT-4o-last (OpenAI et al., 2024a), GLM-4v-Plus (GLM et al., 2024) or Claude3.7-Sonnet (Anthropic, 2025) across a suite of reasoning benchmarks. Further analysis reveals that the performance gap between coupled and decoupled approaches widens as reasoning chains increase in length, confirming the fundamental limitations of end-to-end architectures for extended visual reasoning tasks. The primary contributions of this work are:

- We introduce a novel, training-free framework that decouples reasoning and perception, using a powerful LLM to strategically orchestrate a perception-focused LMM. This plug-and-play approach ensures logical chains remain faithfully anchored to visual evidence, directly mitigating the identified failure mode without costly retraining.

- We demonstrate state-of-the-art performance for open-source models, showing that our approach is remarkably efficient. Our experiments validate that a 39B parameter decoupled model achieves performance parity with, and in some cases surpasses, proprietary giants like GPT-4o and Claude 3.7 Sonnet on a suite of challenging reasoning benchmarks.

- We provide the systematic analysis that quantitatively identifies progressive visual degrounding as a critical failure mode in LMMs during long-chain reasoning. This establishes a foundational understanding of why current end-to-end models often fail in complex, multi-step tasks.

## 2 RELATED WORK

### 2.1 LARGE LANGUAGE MODEL REASONING

The landscape of reasoning within artificial intelligence has been reshaped by the advent of Large Language Models (LLMs). Seminal advancements in this domain originated from text-centric models, where techniques such as Chain-of-Thought (CoT) prompting (Wei et al., 2023; Snell et al., 2024; Wang et al., 2022) marked a pivotal development. CoT and its subsequent elaborations (Lightman et al., 2023; Ning et al., 2023; Qi et al., 2024) enable LLMs to deconstruct complex problems into a sequence of intermediate, manageable steps, thereby emulating a more deliberative human-like reasoning process. This foundational concept has since been architecturally diversified into more structured paradigms, including Program-of-Thoughts (Chen et al., 2022), Table-of-Thoughts (Jin & Lu, 2023), and Tree-of-Thoughts (Yao et al., 2023), each tailored to optimize reasoning pathways for specific problem structures. Further pushing the frontier, recent works have explored advanced learning strategies. OpenAI's O1 (OpenAI et al., 2024b), for instance, integrates reinforcement learning (Ouyang et al., 2022; Schulman et al., 2017; Hu, 2025) with CoT to refine decision-making processes autonomously. Concurrently, models like DeepSeek-R1 (DeepSeek-AI et al., 2025) have pioneered the use of pure reinforcement learning, leveraging algorithms such as Group Relative Policy Optimization (GRPO) (Shao et al., 2024) and rule-based reward systems to cultivate emergent reasoning capabilities without direct supervision.

### 2.2 MULTI-MODAL LARGE LANGUAGE MODEL REASONING

Building upon the successes in the unimodal text domain, research has naturally progressed towards Large Multimodal Models (Zhang et al., 2023; Wu et al., 2025; Rose et al., 2023; Ni et al., 2024). A significant body of work in this area has focused on adapting the established CoT structures for multi-modal contexts (Xu et al., 2025; Thawakar et al., 2025; Sun et al., 2025; Li et al., 2025a) and curating high-fidelity training datasets to elicit these reasoning skills (Du et al., 2025; Shen et al., 2025; See, 2025). For example, Virgo (Du et al., 2025) demonstrated that text-only reasoning data could, to some extent, activate latent multi-modal reasoning abilities. Frameworks such as LLaVA-CoT (Xu et al., 2025) have proposed structured, multi-stage reasoning pipelines, while others like MM-Verify (Sun et al., 2025) have introduced verification mechanisms to enhance the fidelity of the generated reasoning steps.

Despite these advances, **a predominant trend in current multi-modal reasoning research is the direct migration of paradigms initially conceived for text-based tasks.** These approaches, while effective in certain scenarios, often treat visual inputs as mere conditional prompts rather than as a rich source of information that requires deep semantic and spatial processing. Consequently, they often fail to adequately incorporate mechanisms for visual-specific information processing, revealing their inherent limitations when confronted with visually-intensive reasoning tasks that necessitate a profound integration of visual evidence with linguistic logic. **Our work is motivated by this critical gap, aiming to develop a reasoning framework that natively integrates visual and textual modalities from the ground up.**

## 3 METHODOLOGY

In this section, we present our novel training-free framework designed to enhance visual reasoning by maintaining a continuous feedback loop between reasoning and perception. Our approach, structured as a multi-agent system, decouples high-level reasoning from low-level visual perception, enabling a more deliberate and grounded reasoning process.

### 3.1 OVERALL FRAMEWORK

We introduce a novel training-free reasoning framework that continuously incorporates visual information throughout the visual reasoning process, ensuring that the reasoning trajectory remains guided by visual cues at all times. The core idea is to decompose the complex task of visual reasoning into two specialized, collaborative functions: logical deduction and visual data extraction.

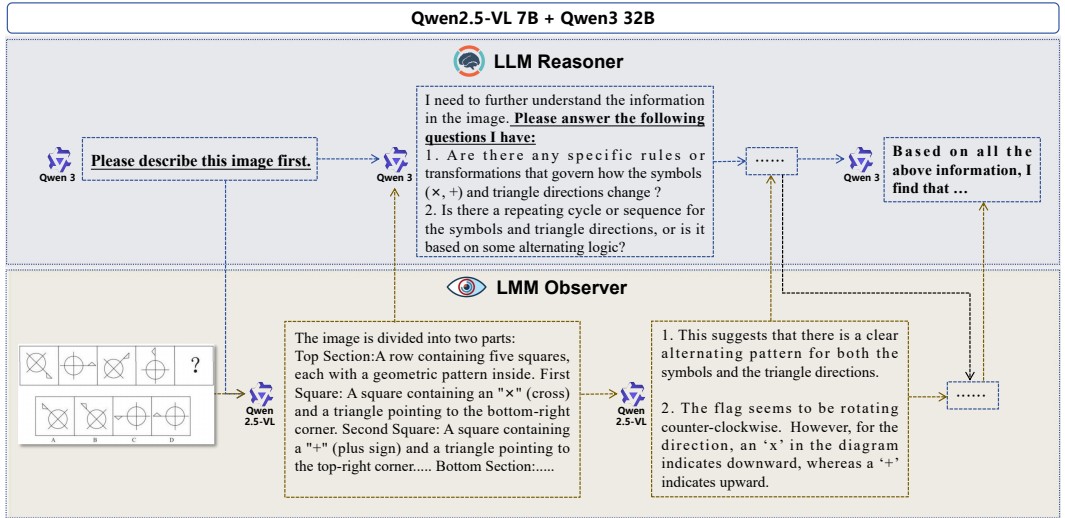

Figure 2: An illustration of our iterative dialogue-based reasoning framework. The framework consists of a non-visual **LLM Reasoner** ($\mathcal{R}$, implemented with Qwen3-32B) and a **LMM Observer** ($\mathcal{P}$, implemented with Qwen2.5-VL 7B). For the given visual reasoning task, $\mathcal{R}$ initiates the dialogue by requesting a general description. Based on the response from $\mathcal{P}$, $\mathcal{R}$ formulates targeted follow-up questions to understand the underlying patterns. This multi-turn, collaborative process enables the LLM Reasoner to deduce the solution by continuously grounding its logic in the visual evidence provided by the LMM Observer.

To this end, our framework consists of two distinct modules: a **LLM Reasoner** ($\mathcal{R}$) and a **LMM Observer** ($\mathcal{P}$). By decoupling reasoning from perception, the framework adopts an iterative, dialogue-based procedure in which the two agents engage in a structured conversation. This conversation is dynamically focused on verifying hypotheses and acquiring visual evidence pertinent to the reasoning path. The LLM Reasoner actively probes the LMM Observer to collect essential visual information, which in turn steers the reasoning process toward a correct and verifiable conclusion. The overall architecture of our framework is illustrated in Figure 2.

### 3.2 MODULE SPECIFICATION

The synergy between the two modules is central to our framework's design. Each module is instantiated with a pre-trained foundation model, requiring no task-specific fine-tuning.

**LLM Reasoner** ($\mathcal{R}$)   This module is implemented using a large language model (LLM). Its primary responsibility is to serve as the cognitive core of the framework. The LLM Reasoner orchestrates the entire problem-solving process. Given an initial user query, it formulates a multi-step reasoning plan. It does not have direct access to the visual input. Instead, it generates a series of targeted, natural language questions ($q_t$ at timestep $t$) to dispatch to the LMM Observer. Based on the returned visual descriptions ($v_t$), it evaluates whether the accumulated information is sufficient to derive a final answer. If not, it generates a follow-up question to probe for more specific details.

**LMM Observer** ($\mathcal{P}$)   This module is instantiated with a Large Multimodal model (LMM). Its sole function is to act as a visual oracle. It receives the image input ($I$) and the textual query ($q_t$) from the LLM Reasoner. Its task is to analyze the visual content of the image and provide a descriptive answer ($v_t$) that directly addresses the query. The LMM Observer's scope is confined to visual understanding; it does not engage in complex reasoning but rather provides the raw perceptual data required by the LLM Reasoner.

## 3.3 ITERATIVE DIALOGUE-BASED REASONING PROCESS

The problem-solving process unfolds as a structured, multi-turn dialogue between $\mathcal{R}$ and $\mathcal{P}$. Let $Q_{user}$ be the initial question posed by the user and $I$ be the input image. The dialogue history at turn $t$ is denoted as $H_t = \{(q_1, v_1), (q_2, v_2), \ldots, (q_{t-1}, v_{t-1})\}$. The process can be formalized as follows:

**Step 1: Initialization (t=1)** The LLM Reasoner $\mathcal{R}$, being non-visual, cannot interpret the image $I$. Its first action is to gain a holistic understanding of the visual scene. It formulates an initial, broad query $q_1$ based on $Q_{user}$. Typically, this query is a generic request for a comprehensive description of the image.

$$q_1 = \mathcal{R}(\text{Prompt}_{\text{init}}, Q_{user}) \tag{1}$$

The LMM Observer $\mathcal{P}$ then processes this query against the image $I$ to produce the first visual description, $v_1$.

$$v_1 = \mathcal{P}(I, q_1) \tag{2}$$

**Step 2: Iterative Reasoning and Probing (t > 1)** For each subsequent step $t$, the LLM Reasoner receives the new visual information $v_{t-1}$ and appends the pair $(q_{t-1}, v_{t-1})$ to the dialogue history $H_t$. With this updated context, $\mathcal{R}$ performs a critical evaluation based on the full dialogue history and the original question:

1. **Sufficiency Check:** It analyzes whether the information contained within $H_t$ is sufficient to conclusively answer $Q_{user}$.
2. **Action Determination:** Based on the check, it decides on one of two actions: ANSWER or QUERY.

If the action is ANSWER, $\mathcal{R}$ synthesizes the information from $H_t$ to generate the final answer $A$, and the process terminates.

$$A = \mathcal{R}(\text{Prompt}_{\text{ans}}, Q_{user}, H_t) \tag{3}$$

If the action is QUERY, it signifies that the existing visual evidence is incomplete. $\mathcal{R}$ then formulates a new, more specific question $q_t$ designed to elicit the missing visual details from $\mathcal{P}$.

$$q_t = \mathcal{R}(\text{Prompt}_{\text{query}}, Q_{user}, H_t) \tag{4}$$

The LMM Observer $\mathcal{P}$ then provides the corresponding visual description $v_t = \mathcal{P}(I, q_t)$, and the loop continues.

This iterative process allows the framework to dynamically construct a detailed understanding of the visual scene, focusing only on the aspects relevant to answering the user's question, thereby mimicking a human-like analytical process. The entire workflow is guided by a set of carefully engineered prompts that define the roles and behavior of the LLM Reasoner, ensuring it adheres to the described dialogue structure.

## 4 EXPERIMENTS

### 4.1 EXPERIMENT SETTINGS

**Implementation details:** We instantiate the LMM Observer with Qwen2.5-VL (Bai et al., 2025) and the LLM Reasoner with Qwen3 (Yang et al., 2025a). Considering both time complexity and the performance of the reasoning framework, we limit the LLM Reasoner to 2 query rounds, with each round allowing up to 3 questions. This constraint reduces time complexity while maintaining excellent performance of the reasoning framework.

**Baseline:** We conduct experiments across a wide range of models, including:

- Open-source unreasoning models: Qwen2.5-VL-3B (Bai et al., 2025), Qwen2.5-VL-7B, Qwen2.5-VL-32B, Qwen2.5-VL-72B, InternVL3-78B (Zhu et al., 2025).
- Open-source reasoning models: VLM-R1-3B-Math (Shen et al., 2025), R1-OneVision (Yang et al., 2025b), OpenVLThinker-7B (Deng et al., 2025), LLaVA-COT (Xu et al., 2025), QVQ-72B-Preview (qvq, 2024).

Table 1: Comprehensive performance comparison on six mathematical and logical reasoning benchmarks. Our models are evaluated against a wide array of state-of-the-art open-source (unreasoning and reasoning-focused) and proprietary LMMs.

| Model | Param (B) | MathVision | MathVerse | WeMath | LogicVista | MM-math | OlympiadBench |
|---|---|---|---|---|---|---|---|
| *Open Source Unreasoning Models* | | | | | | | |
| Qwen2.5-VL-3B | 3 | 21.2 | 29.4 | 21.1 | 28.2 | 22.2 | 7.0 |
| Qwen2.5-VL-7B | 7 | 25.4 | 41.1 | 36.2 | 47.9 | - | - |
| Qwen2.5-VL-72B | 72 | 39.3 | 47.3 | 49.1 | 55.7 | 39.8 | 19.9 |
| InternVL3-78B | 78 | 38.8 | 51 | 46.1 | 55.9 | - | - |
| *Open Source Reasoning Models* | | | | | | | |
| VLM-R1-3B-Math | 3 | 21.9 | 32.2 | 30 | 40.5 | - | - |
| R1-Onevision | 7 | 30.6 | 40.0 | 61.8 | - | - | 17.8 |
| OpenVLThinker-7B | 7 | 25.3 | 47.9 | 64.3 | - | - | 20.1 |
| LLaVA-COT | 11 | - | 22.6 | - | - | - | - |
| QVQ-72B-Preview | 72 | 34.9 | 48.2 | 39 | 58.2 | 22.6 | 20.2 |
| *Proprietary LMMs* | | | | | | | |
| GLM-4v-Plus-20250111 | - | 51.1 | 40.7 | 47.7 | 54.4 | - | - |
| GPT-4o | - | 31.2 | 40.6 | 45.8 | 52.8 | - | - |
| Claude3.7-Sonnet | - | 41.9 | 46.7 | 49.3 | 58.2 | - | - |
| ChatGPT-4o-latest | - | 43.8 | 49.9 | 50.6 | 64.4 | - | - |
| GPT-4.1-20250414 | - | 45.1 | 48.9 | 55.5 | 61.1 | - | - |
| *Ours* | | | | | | | |
| Qwen2.5VL-7B x Qwen3-4B | 11 | 43.2 | 42.6 | 34.7 | 50.8 | 30.2 | 14.7 |
| Qwen2.5VL-7B x Qwen3-32B | 39 | 46.6 | 47.2 | 40.2 | 56.6 | 32.4 | 17.4 |
| Qwen2.5VL-72B x Qwen3-32B | 104 | 52.6 | 54.4 | 51.8 | 60.6 | 42.2 | 22.4 |

- Proprietary LMMs: GLM-4v-Plus (GLM et al., 2024), GPT-4o (OpenAI et al., 2024a), Claude 3.7 Sonnet (Anthropic, 2025), ChatGPT-4o-latest (OpenAI et al., 2024a), GPT-4.1 (OpenAI, 2025).

We compare the above models against our training-free reasoning framework.

**Benchmark:** We adopt six widely used multimodal reasoning benchmarks as our evaluation suite: MathVision (Wang et al., 2024), MathVerse (Zhang et al., 2024), OlympiadBench (He et al., 2024), MM-Math (Sun et al., 2024), WeMath (Qiao et al., 2024), and LogicVista (Xiao et al., 2024). Accuracy (Acc) is reported as the evaluation metric.

## 4.2 MAIN RESULT

We evaluate our proposed framework against a comprehensive suite of state-of-the-art open-source and proprietary models on six challenging visual reasoning benchmarks. The main results are presented in Table 1.

Our framework not only improves upon its base components but also demonstrates highly competitive performance against the leading models in the field. Our premier model, 'Qwen2.5VL-72B x Qwen3-32B', establishes a new state-of-the-art among open-source models on several key benchmarks, achieving top scores on **MathVision (52.6) and MathVerse (54.4)**. Crucially, our framework achieves performance that surpasses several leading proprietary models, despite these models presumably having significantly larger parameter counts and being trained on more extensive datasets.

For instance, on the MathVerse benchmark, our model's score of **54.4** is substantially higher than that of GPT-4o (40.6), Claude 3.7 Sonnet (46.7), and GPT-4.1 (48.9). A similar trend is observed on MathVision, where our score of **52.6** outperforms GLM-4v-Plus (51.1) and all listed GPT and Claude variants. This is a significant finding, suggesting that **a more effective reasoning architecture can compensate for disadvantages in scale, offering a more parameter-efficient path towards advanced visual reasoning.**

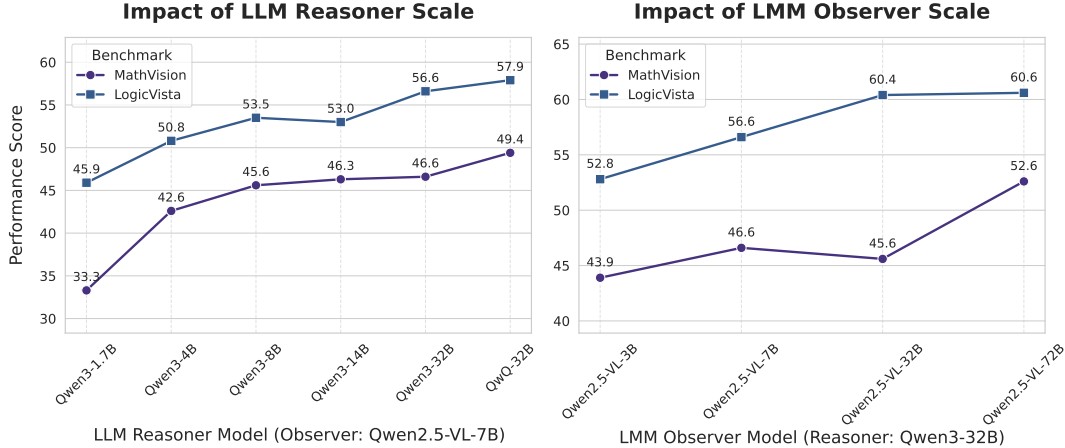

Figure 3: Ablation studies on the impact of component scaling. **Left:** The effect of scaling the LLM Reasoner (from 1.7B to 32B) while keeping the LMM Observer fixed (Qwen2.5-VL-7B). **Right:** The effect of scaling the LMM Observer (from 3B to 72B) while keeping the LLM Reasoner fixed (Qwen3-32B). Both plots show a strong positive correlation between component size and performance on the MathVision and LogicVista benchmarks, indicating that both perceptual and reasoning capabilities benefit from increased model scale.

### 4.3 ABLATION STUDY

#### 4.3.1 IMPACT OF LLM REASONER SCALE

To investigate the influence of the reasoning component's capacity on overall performance, we conducted a series of experiments by varying the scale of the LLM Reasoner while keeping the LMM Observer fixed. We used Qwen2.5-VL-7B as our consistent LMM Observer and scaled the LLM Reasoner across several models from the Qwen3 series, ranging from 1.7B to 32B parameters.

The results, as visualized in the left of Figure 3, demonstrate a clear and positive correlation between the scale of the LLM Reasoner and the model's performance on both the MathVision and LogicVista benchmarks. Specifically, as we increase the parameter count of the reasoner from 1.7B to 32B, the score on MathVision climbs from 33.3 to a peak of 49.4, and the LogicVista score improves from 45.9 to 57.9. **This trend strongly suggests that a larger, more powerful LLM component is crucial for enhancing the model's ability** to handle complex mathematical and logical reasoning tasks that require deep semantic understanding and multi-step thinking. The consistent performance improvement underscores the effectiveness of our architecture, where the LMM Observer grounds the visual information, and a scaled-up LLM Reasoner effectively leverages this information for advanced cognitive tasks.

#### 4.3.2 IMPACT OF LMM OBSERVER SCALE

we also evaluated the impact of the LMM Observer's scale on performance. In this set of experiments, we fixed the LLM Reasoner to a powerful Qwen3-32B model and varied the LMM Observer across the Qwen2.5-VL series, from 3B to 72B parameters.

The results, visualized in right of Figure 3, reveal that the perceptual capabilities of the LMM Observer are equally critical. A distinct upward trend is observed on both benchmarks as the observer's scale increases. On the LogicVista benchmark, performance consistently rises from 52.8 to 60.6, showcasing a strong positive correlation. For MathVision, while there is a slight dip with the 32B model, the largest 72B observer achieves a score of 52.6, a significant leap from the 43.9 obtained with the 3B model.

This analysis highlights that a more capable visual observer provides superior visual grounding and context extraction. A larger LMM Observer can discern more nuanced details from the input images, which are then passed to the LLM Reasoner. This enhanced visual understanding forms a better

foundation for the reasoner to perform complex logical and mathematical deductions, ultimately leading to higher overall accuracy.

### 4.3.3 IMPACT OF DIALOGUE ROUNDS

Table 2: Performance comparison across different numbers of dialogue rounds. Using a fixed Reasoner (Qwen3-32B) and Observer (Qwen2.5-VL-7B).

| LMM Model | LLM Model | round | MathVision | LogicVista | MathVerse |
|---|---|---|---|---|---|
| Qwen2.5-VL-7B | Qwen3-32B | 1 | 41.7 | 55.0 | 48.2 |
| | | 2 | 46.6 | 56.6 | 47.2 |
| | | 3 | 45.1 | 57.0 | 48.7 |
| | | 4 | 42.9 | 56.2 | 46.3 |

A core tenet of our framework is the iterative dialogue between the LLM Reasoner and the LMM Observer. To quantify its impact and identify the optimal level of interaction, we conducted an ablation study detailed in Table 2. For this experiment, we utilized Qwen2.5-VL-7B as the Observer and Qwen3-32B as the Reasoner.

Upon initiating the interactive dialogue (one round), we observe a substantial leap in performance. This improvement continues as the number of rounds increases, typically peaking around the second or third interaction. Interestingly, we observed a slight performance saturation or even a marginal decline when extending the dialogue to four rounds. This suggests that after a certain point, typically three rounds in our experiments, the necessary visual information has been sufficiently extracted. Further interactions may introduce redundant information or noise, leading to less efficient reasoning paths without contributing additional value.

### 4.3.4 IMPACT OF CHAIN-OF-THOUGHT REASONING OF LLM

Table 3: The impact of the LLM Reasoner's Chain-of-Thought (CoT).

| LMM Model | LLM Model | thinking | MathVision | LogicVista | Wemath |
|---|---|---|---|---|---|
| Qwen2.5-VL-3B | Qwen3-32B | Yes | 46.6 | 56.6 | 40.2 |
| | Qwen3-32B | No | 36.8 | 47.0 | 34.5 |
| | Qwen2.5-32B | No | 63.5 | 44.1 | 34.3 |

Our framework is predicated on the idea that a powerful LLM orchestrates the reasoning process. A crucial aspect of this orchestration is the LLM's ability to perform internal reasoning before and during its interaction with the LMM Observer. To validate the importance of this capability, we conducted an experiment to analyze the impact of the LLM Reasoner's Chain-of-Thought (CoT) process. We compare the performance of our standard setup, where the Qwen3-32B reasoner employs CoT ("Yes"), against a variant where its internal reasoning is disabled ("No"), forcing it to generate questions directly.

The results, presented in Table 3, unequivocally demonstrate that the CoT process is indispensable for effective visual reasoning. When CoT is enabled, our model achieves scores of 46.6 on MathVision, 56.6 on LogicVista, and 40.2 on WeMath. In stark contrast, disabling the LLM's reasoning capabilities leads to a dramatic performance collapse across all benchmarks, with scores dropping to 36.8, 47.0, and 34.5, respectively. This represents an average performance degradation of approximately 10 points, highlighting the critical role of the reasoner's internal monologue

### 4.3.5 ANALYZING VISUAL OBJECT COUNT UNDER LONG CHAIN OF THOUGHT

To quantitatively investigate the phenomenon of visual de-grounding in standard LMMs and to validate the efficacy of our proposed framework, we conducted an analysis tracking the introduction

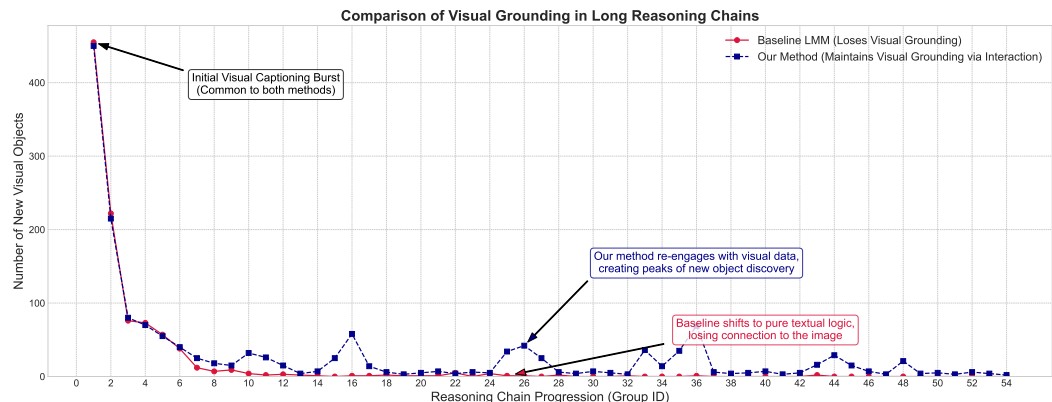

Figure 4: A comparative analysis of visual grounding over long reasoning chains. The baseline LMM (crimson line) rapidly loses its connection to the visual input, with the number of newly introduced visual objects dropping to near zero. This illustrates a shift to purely textual reasoning. In contrast, our proposed framework (blue dashed line) maintains visual grounding by periodically re-engaging with the LMM observer, evidenced by the recurrent spikes in new visual object introduction. This demonstrates our method's ability to produce more faithful, visually-anchored reasoning.

of new visual objects throughout the reasoning process. We define a "new visual object" as an entity mentioned in the reasoning chain that is explicitly tied to the visual content and has not been previously referenced. The reasoning chain is segmented into sequential groups of words, allowing us to measure the rate of new object introduction as the chain extends.

The results of this analysis are presented in Figure 4. As illustrated by the crimson curve, the baseline LMM exhibits a characteristic pattern of severe visual de-grounding. Initially, there is a large burst of new visual objects, corresponding to a high-level captioning or description of the image scene. However, this is followed by a precipitous decline, with the curve rapidly approaching and flatlining at zero. **This trend provides strong quantitative evidence that as the reasoning chain lengthens, the baseline model's process becomes detached from the visual input, transitioning into a purely text-based logical flow** that is vulnerable to hallucination and factual divergence from the image content.

In stark contrast, our proposed method, depicted by the blue dashed curve, demonstrates a fundamentally more robust and faithful reasoning behavior. While it shares a similar initial burst, our framework maintains a persistent connection to the visual context. The periodic spikes observed in the later stages of the reasoning chain are crucial evidence of this capability. These peaks signify moments where our LLM Reasoner strategically determines a need for further visual evidence and initiates a query to the LMM Observer.

## 5 CONCLUSION

In this work, we identify and address the critical failure mode of progressive visual de-grounding in Large Multimodal Models. We demonstrate that as reasoning chains extend, standard LMMs detach from visual evidence, leading to flawed, text-driven conclusions. To solve this, we introduce a novel, training-free framework that decouples reasoning from perception. Our plug-and-play pipeline uses a powerful LLM Reasoner to strategically interrogate a perception-focused LMM Observer, ensuring that the inference process remains faithfully anchored to the visual input. Our experiments validate this approach, showing that a 39B parameter open-source model can achieve performance parity with colossal proprietary systems like GPT-4o. Our findings confirm that decoupling is a highly effective and resource-efficient path toward building more reliable and transparent multimodal reasoning systems.

ETHICS STATEMENT

This work presents a training-free visual reasoning framework built upon publicly available Large Multimodal Models (LMMs) and openly released multimodal reasoning benchmarks. All datasets used in our evaluation are obtained from public sources that provide licenses or terms of use permitting research. No new data collection involving human participants is conducted in this study. The framework itself does not store, transmit, or process any personally identifiable information. We caution that, as with other LMMs, our method may inherit limitations or biases present in the underlying pretrained models and benchmarks. Potential misuse of the framework, such as application in high-stakes decision-making without human oversight, may lead to unintended consequences. We encourage responsible and transparent use of the framework and recommend that outputs be critically reviewed by human experts in sensitive applications.

REPRODUCIBILITY STATEMENT

To ensure reproducibility, we release our complete source code, experiment scripts, and configuration files in the supplementary material (decouple.zip). The package includes documentation specifying software dependencies, environment setup, and commands for executing all experiments. Using these resources, readers can exactly replicate the results presented in this paper.

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

# A APPENDIX

## A.1 USE OF LLMs

Large language models (LLMs) are used solely to assist in the preparation of this manuscript. They help improve the clarity, coherence, and conciseness of the text, refine phrasing, and ensure that the language conforms to academic writing standards. All conceptual content, experimental design, data analysis, and conclusions are developed entirely by the authors.

## A.2 CASE STUDY

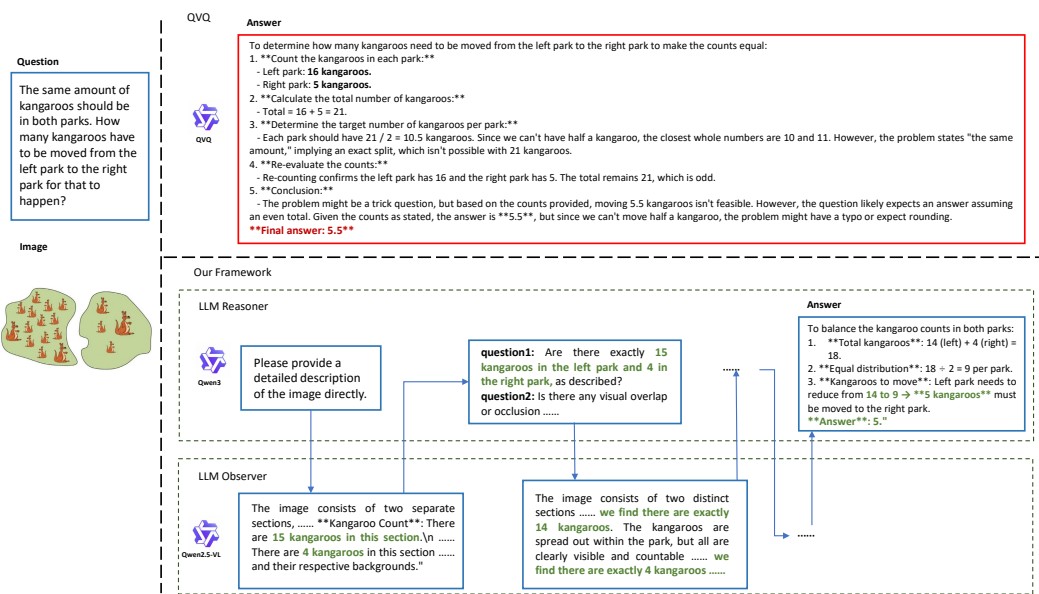

Figure 5: A Comparative Case Study: The QvQ-32B Model vs. Our Proposed Framework

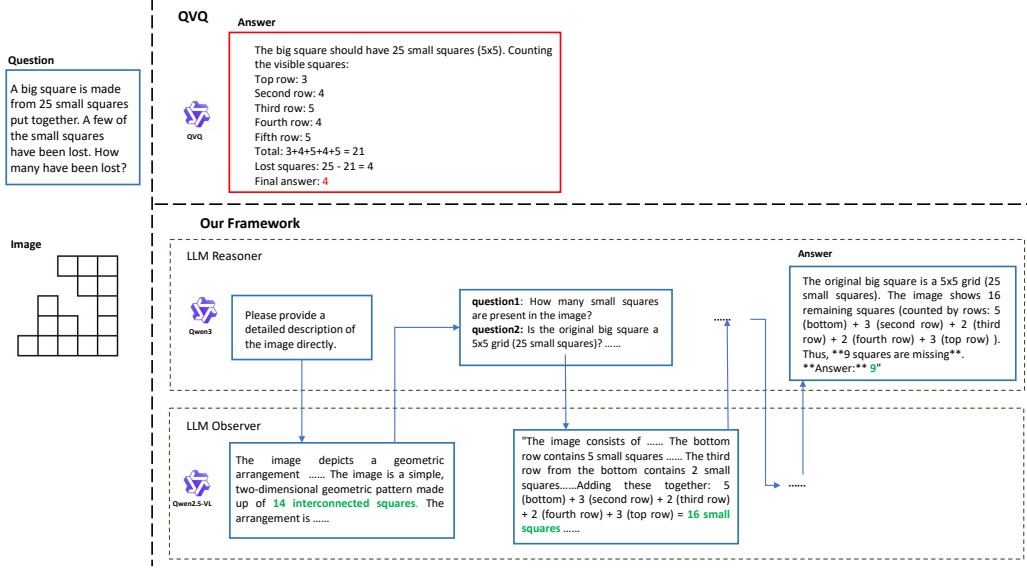

Figure 6: A Comparative Case Study: The QvQ-32B Model vs. Our Proposed Framework

As shwon in Figure 5, this case study demonstrates the robustness of our decoupled framework compared to a traditional end-to-end reasoning LMMs on a visual reasoning task that hinges on precise visual perception. The problem requires the model to determine how many kangaroos must be moved to equalize the count in two separate parks. It performs an initial visual captioning to extract a textual description of the scene, but critically, its subsequent reasoning relies exclusively on this static description without ever revisiting the image to confirm or revise its initial perceptions. This makes the entire inference chain brittle; an early perceptual error, such as miscounting the kangaroos, is locked in and inevitably propagates, guaranteeing a flawed conclusion. In stark contrast, our framework succeeds by implementing a dynamic, iterative dialogue between its two specialized components. The LLM Reasoner does not passively accept the initial visual assessment. Instead, it actively cross-examines the LMM Observer through multiple rounds of targeted queries. This interrogative process compels the Observer to repeatedly re-engage with the visual information, allowing it to identify and correct initial misperceptions (e.g., refining the kangaroo count from 15 to a more accurate 14). This continuous verification loop ensures that the final reasoning is constructed upon a foundation of accurate, grounded visual facts, leading to a robust and correct outcome.

