# OpenReview forum: "Decoupling Reasoning and Perception: An LLM-LMM Framework for Faithful Visual Reasoning"
_ICLR.cc/2026/Conference — ICLR 2026 Conference Withdrawn Submission_

### Official Review · Reviewer_Hrox · 2025-10-24

**Soundness:** 2
**Presentation:** 1
**Contribution:** 1
**Rating:** 2
**Confidence:** 5

**Summary:**

The paper proposes a training-free pipeline that decouples reasoning (handled by a large language model, LLM) and perception (handled by a large multimodal model, LMM) for visual reasoning tasks. The idea: the Reasoner (LLM) probes the Observer (LMM) via questions to extract visual information, avoiding the long‐chain hallucination problem in end-to-end LMMs. Experiments on multiple visual reasoning benchmarks claim state-of-the‐art results among open-source models.

**Strengths:**

1. The framework is conceptually simple and does not require fine-tuning or architecture modification, making it easy to integrate with existing LLMs and LMMs. This training-free plug-and-play design provides a practical advantage, particularly for real-world applications where compute and retraining budgets are limited.
2. Extensive experiments are conducted on multiple visual reasoning benchmarks (e.g., MathVision, LogicVista, MMMU-Pro), showing consistent improvements across diverse reasoning domains.

**Weaknesses:**

1. Novelty and Significance Are Limited. The decoupling of perception and reasoning is intuitive and has been explored in various “multi-agent” or “modular” multimodal reasoning works. The paper claims “training-free plug-and-play” as main novelty, but that alone is not a strong enough methodological advance for ICLR. While the authors claim a “progressive visual de-grounding” failure mode (where an LMM loses visual grounding as reasoning chains extend) and propose to address it, the novelty of diagnosing this failure and the remedy is not compellingly distinguished from prior modular reasoning or agent-based VLM frameworks.
2. Methodological / Theoretical Depth Is Insufficient. The framework is described in algorithmic form, but lacks deeper theoretical analysis of why the decoupled reasoning + perception architecture fundamentally improves reasoning fidelity. For instance: under what conditions does iterative questioning guarantee better grounding? What are failure modes (e.g., the Observer gives wrong answers, or the Reasoner asks sub-optimal questions)?The paper is largely empirical; the reasoning module is essentially an LLM prompt policy, and the perception module is a standard LMM. The structural change is light, which may reduce the interest for a top-tier conference that prizes stronger algorithmic contributions.
3. Although Table 1 reports large improvements over other open-source and proprietary models, the experimental fairness needs closer inspection: Are both baseline and proposed systems evaluated under the same constraints (inference time, number of reasoning steps, compute budget, prompt design)? The paper lacks clarity on cost/latency trade-offs with the iterative questioning framework. The claim that a 39B parameter model “achieves parity” with proprietary systems may be overstated: differences in dataset, prompt tuning, model access, unseen training data may confound the comparison. Without explicit ablation on compute/latency/energy, the practical significance is unclear. The evaluation appears concentrated on a specific subset of reasoning benchmarks (MathVision, LogicVista, etc.). There is limited evidence of broader applicability (e.g., real‐world vision-language tasks, non‐math reasoning, generalization to unseen domains). That limits generalization and impact.

**Questions:**

1. How does your proposed “decoupling” differ fundamentally from prior modular or agent-based reasoning frameworks (e.g., ReAct, ViperGPT, or recent multi-agent VLM architectures)? If it's just component replacement or simple process standardization, is the innovation sufficient?
2. Under what conditions can iterative LLM–LMM questioning guarantee improved grounding or reasoning accuracy? Can you formalize any theoretical insight or expectation?
3. Is there a clear decision policy for how many questioning rounds are used, or when to terminate the interaction?
4. How is information consistency maintained between LMM answers and LLM reasoning memory?
5. Have you tested robustness on more diverse datasets — e.g., non-math or real-world visual tasks — to validate generality?
6. Could performance gains come merely from additional context length or multi-turn reasoning rather than true “decoupling”?

---

### Official Review · Reviewer_hrRV · 2025-10-28

**Soundness:** 3
**Presentation:** 3
**Contribution:** 3
**Rating:** 4
**Confidence:** 4

**Summary:**

The paper proposes a training-free, plug-and-play framework that decouples reasoning and perception for visual reasoning. A text-only LLM Reasoner conducts multi-step planning and questioning, while a LMM Observer answers targeted, image-grounded queries. The agents interact in an iterative dialogue until the Reasoner judges the evidence sufficient to answer. On six visual-reasoning benchmarks (e.g., MathVision, MathVerse, LogicVista), the decoupled pipeline (e.g., Qwen2.5-VL-7B × Qwen3-32B) outperforms its coupled baselines and approaches or exceeds several proprietary systems.

**Strengths:**

1. Clear, simple decoupling of roles—LLM for logic, LMM for perception—implemented as dialogue-style querying without training. This is a crisp alternative to end-to-end “reason-while-seeing” pipelines and directly targets loss of visual grounding.
2. Empirical study is broad: six benchmarks, comparisons to open-source and proprietary LMMs, and multiple ablations (scale of Reasoner/Observer, #rounds, Reasoner CoT).
3. Results suggest architecture matters: a modest-sized, decoupled open-source stack can match or beat larger proprietary baselines on several visual reasoning tasks, pointing to a parameter-efficient path forward.

**Weaknesses:**

1. The method’s practicality hinges on multi-turn calls. The paper limits the Reasoner to “2 rounds × up to 3 questions,” but there is no wall-clock or token-cost analysis vs. coupled baselines. Real-world systems need expected latency per example, variance, and accuracy-vs-cost trade-offs under tighter query budgets.
2. The Observer is said to be “perception-only,” but LMMs often sneak in reasoning. How is prompting enforced to prevent the Observer from performing long-form logic or speculative hallucinations? What stops information leakage (e.g., Reasoner smuggling assumptions into questions)?
3. The related work focuses on CoT-style methods; however, comparisons to verification-augmented or tool-enabled LMMs (e.g., visual verification modules, external OCR/detection tools) are missing. This matters because a strong verifier might reduce drift without multi-turn agent interaction.

**Questions:**

1. Could you include a qualitative taxonomy of errors (e.g., wrong question, misread attribute, over-generalization) with case studies illustrating when the decoupled scheme still fails?
2. Could you use proprietary LMMs as reasoners/observers, and make more combinations (reasoner/observer) like  InternVL3 x Qwen.

---

### Official Review · Reviewer_93UP · 2025-10-28

**Soundness:** 4
**Presentation:** 3
**Contribution:** 1
**Rating:** 2
**Confidence:** 4

**Summary:**

In this paper, the authors notice that LMMs increasingly rely on textual logic as their reasoning chains extend, progressively losing grounding in the underlying visual information. Based on this observation, the authors propose a training-free visual-reasoning pipeline, prompting a LLM and a LMM to achieve better reasoning.

**Strengths:**

1. The experiments in the paper are very comprehensive. The authors conduct extensive ablation studies, evaluating models of different scales, varying numbers of reasoning rounds and the use of CoT prompting.

2. The authors demonstrate through Figure 4 that their method indeed leads to increased attention to the visual content.

3. The related work section is thorough and covers most of the mainstream MLLM studies.

4. The overall writing is clear and easy to follow.

5. The code is open source.

**Weaknesses:**

Given that the motivation, literature review, and experiments in this paper are sound, my main concern lies in the novelty of the proposed method. The authors attempt to introduce a multi-round prompting approach that leverages the reasoning capabilities of LLMs to enhance the reasoning performance of MLLMs. The claimed contributions can be summarized as twofold:

1. identifying that LLMs excel at textual reasoning while MLLMs are better at visual analysis, and therefore proposing to combine the two;

2. attempting to further improve reasoning ability through multi-round interactions.

However, both contributions lack sufficient originality. The first point was already explored in prior published work [1] about a year ago, and the second point has been extensively studied in recent agent-related research, many of which also involve multi-turn interactions. Some studies have even begun exploring the automated design of such pipelines.

In summary, while the paper is technically solid, I believe it does not meet the level of originality expected.

[1] Yuxuan Qiao, Haodong Duan, Xinyu Fang, Junming Yang, Lin Chen, Songyang Zhang, Jiaqi Wang, Dahua Lin, & Kai Chen. (2024). Prism: A Framework for Decoupling and Assessing the Capabilities of VLMs.

**Questions:**

1. I hope the authors can provide a more detailed explanation of their novelty.

2. In Table 1, it would be helpful to include some more recent MLLMs, such as OpenAI’s o3 model. Of course, it is understandable if the proposed method performs worse than o3; however, including such models would provide a valuable reference point against current state-of-the-art MLLMs, allowing readers to better understand the effectiveness of the proposed approach.

---

### Official Review · Reviewer_WfEu · 2025-10-30

**Soundness:** 2
**Presentation:** 3
**Contribution:** 2
**Rating:** 4
**Confidence:** 4

**Summary:**

This paper addresses the loss of grounded reasoning over long chains of thought in MLLMs in vision-language tasks. To fix this issue, the authors propose a training-free framework that uses two parallel models altogether - one MLLM to obverse the image and the question, and another LLM to perform the reasoning and provide controlled guidance for the MLLM. The authors show this approach have shown superior performance over proprietary models like GPT4.1 on multiple benchmarks such as MathVision and MathVerse.

**Strengths:**

I personally find the work possesses the following highlights:

1. The method is easy to follow and intuitive to understand.
2. The original problem, i.e. the loss of visual grounding over long CoT,  is well demonstrated via the analyses in the manuscript, in particular those in the ablation study part.

**Weaknesses:**

Although I appreciate the simplistic training-free hack for improvements on VL CoT tasks, unfortunately I find several aspects of this paper need to be further scrutinized upon:

1. **A major flaw in the proposed method design is a lack of quantitative measure for terminating the iterative process.** In Chap 3.3, the authors introduce the ANSWER/QUERY action. However, this design of the ‘stop signal’ is purely qualitative and empirical. In theory, such an iterative process of output/input exchange between the Reasoner and the Observer could still go on as long as forever. Furthermore, as suggested in Chap 4.3.3, the authors check the impact of dialogue rounds, which makes me wonder if the length of the CoT in the proposed approach is indeed configured as a pre-determined hyperparameter. In all, have the authors deployed any quantitative metrics in the process, in order to show that their framework can genuinely figure out when to stop reasoning in a quantitative manner? For the time being, unfortunately, I’m not seeing any concrete evidence for that.

2. **The current framework’s setup is apparently Qwen-biased.** The authors only use variants of Qwen as their backbone models. This could potentially risk having the results be skewed towards the pre-training data that are used to construct Qwen. I believe the framework can be extended to any applicable open source backbone MLLMs and LLMs such as variants of Intern/InternVL and LLaMA/LLaVA to mitigate potential contextual bias introduced by the pre-trained component models themselves.

3. **Potentially unleveled comparisons are present in Table 1.** The proposed framework uses two models in combination. Thus I believe it’s unfair to compare a larger proposed model against any standalone baseline model that is way smaller in parameter size. For example, the 11B proposed model (Qwen2.5VL-7B x Qwen3-4B) should only be compared with those MLLMs/LLMs that are on the same 10B scale.

**Questions:**

Please find my main concerns in the Weaknesses section.

---

### Note · Authors · 2025-11-12

**Comment:**

After careful consideration, we request withdrawal of manuscript "Decoupling Reasoning and Perception: An LLM-LMM Framework for Faithful Visual Reasoning".
We deeply regret the time invested by reviewers and editors. We value your conference and hope to submit future work.

**Withdrawal Confirmation:**

I have read and agree with the venue's withdrawal policy on behalf of myself and my co-authors.